# The anti-PD-L1/CTLA-4 bispecific antibody KN046 in combination with nab-paclitaxel in first-line treatment of metastatic triple-negative breast cancer: a multicenter phase II trial

Qiao Li [1], Jiaxuan Liu[1], Qingyuan Zhang[2], Quchang Ouyang[3], Yang Zhang[4], Qiang Liu [5], Tao Sun[6], Feng Ye [7], Baochun Zhang[8], Summer Xia[9], Bangyong Zhang[9] & Binghe Xu[10] ✉

This multicenter, phase II study (NCT03872791) aims to evaluate the efficacy and safety of the anti-PD-L1/CTLA-4 bispecific antibody KN046 combined with nab-paclitaxel in the first-line treatment of patients with metastatic triple-negative breast cancer (TNBC). The primary endpoints included objective response rate (ORR) and duration of response (DoR), and secondary endpoints included progression-free survival (PFS), overall survival (OS) rate, safety, and the correlation of PD-L1 status with clinical efficacy. This trial met pre-specified endpoints. 27 female patients were enrolled sequentially to receive KN046 in two dose levels (3 mg/kg or 5 mg/kg). Among the 25 evaluable patients, the ORR achieved 44.0% (95% CI, 24.4% − 65.1%), and the median DoR was not mature. The median PFS reached 7.33 months (95%CI, 3.68 − 11.07 months), and the median OS was 30.92 months (95%CI, 14.75 - NE months). In PD-L1 positive patients, PFS was 8.61 months (versus 4.73 months) and the 2-year OS rate was 62.5% (versus 57.1%) compared to PD-L1 negative patients. Patients tolerated well the combination therapy. In general, KN046 combined with nab-paclitaxel showed favorable efficacy and survival benefits with tolerable toxicity in the first-line treatment of metastatic TNBC, especially PD-L1 positive, which is worth further investigation.

Breast cancer is the most common malignancy in women worldwide. In China, it is estimated that in 2020, there were 416,000 newly diagnosed breast cancer patients, accounting for 18.4% of the world's new cases of breast cancer[1]. Triple-negative breast cancer (TNBC) is a subtype of breast cancer in which both estrogen receptor (ER) and progesterone receptor (PR) expression are <1% and human epidermal growth factor receptor 2 (HER2) is negative, accounting for 15%-20% of all breast cancers. TNBC is characterized by higher risk of early recurrence, higher frequency of metastasis, and worse prognosis. The median overall survival (OS) of metastatic TNBC is about 8–15 months and the median progression-free survival (PFS) with first-line standard chemotherapy treatment is about 5.6 months according to statistics. Due to the lack of effective molecular therapeutic targets, conventional chemotherapy is still the dominant treatment option for TNBC,

but most patients quickly develop resistance to chemotherapy and then relapse and metastasis. It is necessary to develop new therapeutic strategies for metastatic TNBC.

In recent years, the consensus agrees resoundingly that TNBC is a highly heterogeneous disease, and due to the immunogenic nature different from other subtypes of breast cancer, immune checkpoint inhibitor (ICI) has emerged as the most promising treatment strategy for long-term survival benefits among patients with this intractable subtype of breast cancer. TNBC with higher programmed death ligand-1 (PD-L1) expression, higher tumor mutation burden (TMB), and higher tumor-filtrating lymphocytes (TILs) have been demonstrated to benefit from the treatment of ICI[2]. Despite recent FDA (Food and Drug Administration) approval of ICIs atezolizumab and pembrolizumab ushered in a new era of treatment of advanced TNBC, the overall survival benefit of these patients remains modest[3–8], which creates the necessity to develop new therapeutic strategies.

KN046 is a recombinant humanized anti-PD-L1/cytotoxic T lymphocyte antigen-4 (CTLA-4) bispecific single-domain antibody-Fc fusion protein. It simultaneously blocks CTLA-4 with CD80/85 and PD-L1 with programmed cell death-1 (PD-1) to inhibit immunosuppressive effects, which restores T-cell effector immune response to tumor and deletes $T_{reg}$ cells (suppress tumor immunity) in tumor

microenvironment. Preclinical studies have demonstrated that KN046 has a better effect on T cell activation than PD-(L)1 monotherapy or the combination of PD-(L)1 and CTLA-4 monotherapy. The limited peripheral distribution reduces treatment-associated on-target off-tumor toxicity, and the preclinical study has demonstrated that the toxicity of KN046 is lower than that of the anti-CTLA-4 inhibitor. The first-in-human study has demonstrated that KN046 monotherapy has an acceptable safety profile and is in line with previously reported safety data from other ICIs, and the preliminary efficacy results are promising[9]. Several clinical trials of KN046 monotherapy and combination therapy were conducted in different advanced solid tumors, including non-small cell lung cancer (NSCLC), hepatocellular carcinoma (HCC), pancreatic cancer, esophageal squamous cell carcinoma, and rare thoracic tumors, which have shown promising efficacy and well-tolerated toxicity[10–14].

In this work, we conduct the present multicenter, open-label, phase II trial to evaluate the efficacy and safety of anti-PD-L1/CTLA-4 bispecific antibody KN046 in combination with nab-paclitaxel in the first-line treatment of locally advanced inoperable or metastatic TNBC patients regardless of PD-L1 status. The combination therapy shows favorable clinical efficacy and encouraging survival outcomes, especially for PD-L1 positive patients, and the safety profile is manageable. The 3 mg/kg Q2W dose level of KN046 in combination with nab-paclitaxel is associated with comparable efficacy and superior safety, which is worth evaluating in future trials.

## Results

### Patients

From May 30, 2019 to June 30, 2020, 27 eligible patients with treatment-naïve locally advanced inoperable or metastatic TNBC were enrolled. Sixteen patients were assigned to receive KN046 3 mg/kg Q2W plus nab-paclitaxel (dose level [DL]-1 group), and the other 11 patients were assigned to receive KN046 5 mg/kg Q2W plus nab-paclitaxel (DL-2 groups) sequentially, following a dose escalation part and a dose expansion part afterward.

The clinical cutoff date for PFS and OS analyses was on Aug 21, 2022. The median age of all eligible patients was 50 years old, and 100% were female. 15 (55.6%) patients had been treated with neoadjuvant or adjuvant taxanes and anthracycline chemotherapy. Out of 27, 9 (33.3%) patients were PD-L1 positive (PD-L1 expression ≥1%), and 15 (55.6%) patients were PD-L1 negative (PD-L1 expression <1%). Besides, 3 of 27 (11.1%) patients had an unknown PD-L1 status because of histological sample detachment. The detailed baseline characteristics of enrolled patients at the two dose levels groups were presented in Table 1.

### Efficacy outcomes

By the clinical cutoff date (Aug 21, 2022), the median follow-up was 32.0 months (interquartile range [IQR]: 29.1–36.1). The best overall tumor response presented in Table 2 was assessed by the Independent Review Committee (IRC) based on the intention-to-treat (ITT) population; 2 patients withdrew from the trial before the first tumor assessment and were excluded from the evaluable population. Among the 25 evaluable patients, the objective response rate (ORR) achieved 44.0% (95% CI: 24.4–65.1%), the disease control rate (DCR) achieved 96.0% (95% CI: 79.7–99.9%), and the clinical benefit rate (CBR) achieved 52.0% (95% CI: 31.1–72.2%). The median duration of response (DoR) has not yet been reached. In the DL-1 group with 3 mg/kg Q2W of KN046, the ORR achieved 60.0% (95% CI: 32.3–83.7%), the DCR achieved 100.0% (95% CI: 78.2–100.0%), and the CBR achieved 60.0% (95% CI: 32.3–83.7%). In the DL-2 group with 5 mg/kg Q2W of KN046, the ORR achieved 20.0% (95% CI: 2.5–55.6%), the DCR achieved 90.9% (95% CI: 55.5–99.8%), and the CBR achieved 40.0% (95% CI: 12.2–73.8%). The best overall tumor response presented in Supplementary Table 1 was assessed by investigators based on the intention-to-treat (ITT) population. Among the 25 evaluable patients, the ORR achieved 36.0%

## Table 1 | Baseline patient characteristics

| Characteristics | KN046 3 mg/kg Q2W + nab-paclitaxel (n = 16) | KN046 5 mg/kg Q2W + nab-paclitaxel (n = 11) | Total (n = 27) |
|---|---|---|---|
| Age, median (range) 1 | 53.5 (35–70) | 45 (33–62) | 50 (33–70) |
| *ECOG PS* | | | |
| 0 | 9 (56.3%) | 5 (45.5%) | 14 (51.9%) |
| 1 | 7 (43.8%) | 6 (54.5%) | 13 (48.1%) |
| *Grade* | | | |
| 1 | 0 | 0 | 0 |
| 2 | 4 (25.0%) | 2 (18.2%) | 6 (22.2%) |
| 3 | 7 (43.8%) | 6 (54.5%) | 13 (48.1%) |
| UNK | 5 (31.3%) | 3 (27.3%) | 8 (29.6%) |
| *Stage (cTNM)* | | | |
| IIIa | 1 (6.3%) | 0 | 1 (3.7%) |
| IIIb | 0 | 1 (9.1%) | 1 (3.7%) |
| IIIc | 2 (12.5%) | 0 | 2 (7.4%) |
| IV | 13 (81.25%) | 10 (90.9%) | 23 (85.2%) |
| *Distant metastasis* | | | |
| Yes | 14 (87.5%) | 9 (81.8%) | 23 (85.2%) |
| No | 2 (12.5%) | 2 (18.2%) | 4 (14.8%) |
| *Number of disease sites* | | | |
| 0–3 | 3 (18.8%) | 3 (27.3%) | 6 (22.2%) |
| ≥4 | 13 (81.3%) | 8 (72.7%) | 21 (77.8%) |
| *Metastatic site* | | | |
| Lymph nodes | 10 (62.5%) | 6 (54.5%) | 16 (59.3%) |
| Lung | 7 (43.8%) | 6 (54.5%) | 13 (48.1%) |
| Liver | 5 (31.3%) | 1 (9.1%) | 6 (22.2%) |
| Brain | 1 (6.3%) | 1 (9.1%) | 2 (7.4%) |
| Bone | 5 (31.3%) | 4 (36.4%) | 9 (33.3%) |
| *PD-L1 status* | | | |
| <1% | 7 (43.8%) | 8 (72.7%) | 15 (55.6%) |
| ≥1% | 6 (37.5%) | 3 (27.3%) | 9 (33.3%) |
| UNK | 3 (18.8%) | 0 | 3 (11.1%) |

Data are presented as n (%).
*ECOG PS* Eastern Cooperative Oncology Group performance status, *UNK* unknown, *PD-L1* programmed death-ligand 1.

**Table 2 | Best overall response assessed by IRC**

| | KN046 3 mg/kg Q2W + nab-paclitaxel (n = 15) | KN046 5 mg/kg Q2W + nab-paclitaxel (n = 10) | Total[a] (n = 25) |
|---|---|---|---|
| *Best overall response, n (%)* | | | |
| Complete response, CR | 1 (6.7%) | 0 | 1 (4.0%) |
| Partial response, PR | 8 (53.3%) | 2 (20.0%) | 10 (40.0%) |
| Stable disease, SD | 6 (40.0%) | 7 (70.0%) | 13 (52.0%) |
| Progressive disease, PD | 0 | 1 (10.0%) | 1 (4.0%) |
| Objective response rate, ORR (95% CI) [CR + PR] | 60.0% (32.3–83.7%) | 20.0% (2.5–55.6%) | 44.0% (24.4–65.1%) |
| Disease control rate, DCR (95% CI) [CR + PR + SD] | 100.0% (78.2–100.0%) | 90.0% (55.5–99.8%) | 96.0% (79.7–99.9%) |
| Clinical benefit rate, CBR (95% CI) [CR + PR + SD ≥ 24 weeks] | 60.0% (32.3–83.7%) | 40.0% (12.2–73.8%) | 52.0% (31.3–72.2%) |
| Median DoR | Not reached | Not reached | Not reached |

[a]Two patients withdrew from the trial before the first tumor assessment and were excluded from the evaluable population.

*IRC* independent review committee.

(95% CI: 18.0–57.5%), the DCR achieved 96.0% (95% CI: 79.7–99.9%), and the CBR achieved 52.0% (95% CI: 31.1–72.2%). The median DoR reached 11.93 months (95% CI: 5.59–NE months).

## Survival outcomes

By the clinical cutoff date (Aug 21, 2022), 12 (48.0%) patients in the ITT population had died, including 7 (46.7%) patients in the DL-1 group and 5 (50.0%) patients in the DL-2 group.

The median PFS in the ITT population was 7.33 months (95% CI: 3.68–11.07 months). Among the patients with PD-L1 expression ≥ 1%, the median PFS was 8.61 months (95% CI: 1.61–NE months), and the median PFS was 4.73 months (95% CI: 3.61–11.07 months) in patients with PD-L1 expression <1%. There is no statistically significant difference in PFS based on the PD-L1 expression status in the ITT population (Fig. 1A, B and Table 3).

The median OS in the ITT population was 30.92 months (95% CI: 14.75–NE months), the 1-year OS rate was 73.9% (95% CI: 50.9–87.3%) and the 2-year OS rate was 60.1% (95% CI: 37.2–76.9%). Among the patients with PD-L1 expression ≥ 1%, the median OS was 26.14 months (95% CI: 8.61–NE months) and the 2-year OS rate was 62.5% (95% CI: 22.9–86.1%). The median OS was 30.92 months (95% CI: 6.01–NE months), and the 2-year OS rate was 57.1% (95% CI: 25.4–79.6%) among the patients with PD-L1 expression <1%. OS did not show statistical significance based on the PD-L1 expression status in the ITT population, and both PD-L1 negative and positive patients derived OS benefits from the therapy (Fig. 1C, D and Table 3).

The survival outcomes between the two dose groups were reported (Supplementary Table 2). In the DL-1 group with 3 mg/kg Q2W of KN046, the median PFS was 8.61 months (95% CI: 3.71–NE months), and the median OS was not reached (95% CI: 8.61–NE months). In the DL-2 group with 5 mg/kg Q2W of KN046, the median PFS was 3.65 months (95% CI: 1.61–9.10 months), and the median OS was 27.73 months (95% CI: 6.01–NE months). The survival outcomes comparing PD-L1 expression levels between the two groups were also presented in Supplementary Table 2.

## Safety and tolerability

In total, treatment-emergent adverse events (TEAEs) were reported in all evaluable patients (*n* = 27) and were all associated with treatment. The incidence of ≥Grade 3 TEAEs was 66.7% (18/27), and the incidence of serious adverse events (SAEs) was 33.3% (9/27). TEAEs leading to study drug withdrawal were reported in 9 (33.3%) patients. By the clinical cutoff date (Aug 21, 2022), TEAEs leading to death were reported in 2 (7.4%) patients; one was due to pancreatitis, and the other was due to disease progression. There was no TEAE leading to death associated with KN046 (Table 4).

The most reported treatment-related adverse events (TRAEs) in the ITT population were neutropenia, leukopenia, alopecia, elevated alanine aminotransaminase (ALT) and elevated aspartate amino-transferase (AST) (Table 5). The most reported ≥Grade 3 TRAEs were neutropenia, leukopenia, elevated γ-glutamyl transferase (γ-GGT), hypokalemia, and elevated AST (Table 5). The majority of ≥Grade 3 TRAEs were hepatotoxicity and hematotoxicity, which were reversible after symptomatic and supportive treatment.

Immune-related adverse events (irAEs) were reported in 13 (48.1%) patients. The most reported irAEs were hypothyroidism (3 patients, 11.1%), infusion-related reaction (2 patients, 7.4%) and immune-mediated liver disease (2 patients, 7.4%). Most irAEs were Grade 1 or 2. Grade ≥3 irAEs and immune-related SAEs were only reported in the DL-2 group with 5 mg/kg of KN046. Grade ≥3 irAEs were reported in 3 (11.1%) patients, 2 patients with Grade 3 immune-mediated liver disease, and 1 patient with Grade 3 rash. Immune-related SAEs were reported in 1 (3.7%) patient with immune-mediated liver disease (Table 4; Supplementary Table 3).

## Discussion

Immunotherapy has revolutionized the development of oncology, and ceaseless breakthroughs in ICIs bring great hope to cancer patients. Due to the limited clinical efficacy of ICI monotherapy, the combination treatment of CTLA-4 and PD-L1/PD-1 blockers was then evaluated to improve the response to immunotherapy and achieved promising results. The classic combination of ipilimumab (anti-CTLA-4) plus nivolumab (anti-PD-1) has shown clinically meaningful successes in a variety of advanced solid tumors, including melanoma, NSCLC, HCC, renal cell carcinoma, and colorectal cancer[15]. However, the risks of toxicities increase accordingly; thus, the broad application of this combination has been limited. KN046, a recombinant humanized anti-PD-L1/CTLA-4 bispecific antibody, was designed to achieve the synergistic effect of simultaneous blockade of PD-L1/PD-1 and CTLA-4 pathways. Preclinical and clinical studies have shown promising efficacy and tolerable toxicities in the advanced treatment of a variety of solid tumors. The phase III pivotal study (ENREACH-PDAC-01) has been conducted to verify the efficacy and safety of KN046 plus nab-paclitaxel and gemcitabine in the first-line treatment of advanced pancreatic cancer[16]. The first interim analysis of another phase III clinical study (ENREACH-LUNG-01) has announced that KN046, in combination with platinum-containing chemotherapy in the first-line treatment of advanced unresectable or metastatic squamous NSCLC, significantly extended the PFS of ITT patients and has met its prespecified endpoint[17]. The application prospect of KN046 in immunotherapy of advanced tumors is worthy of expectation.

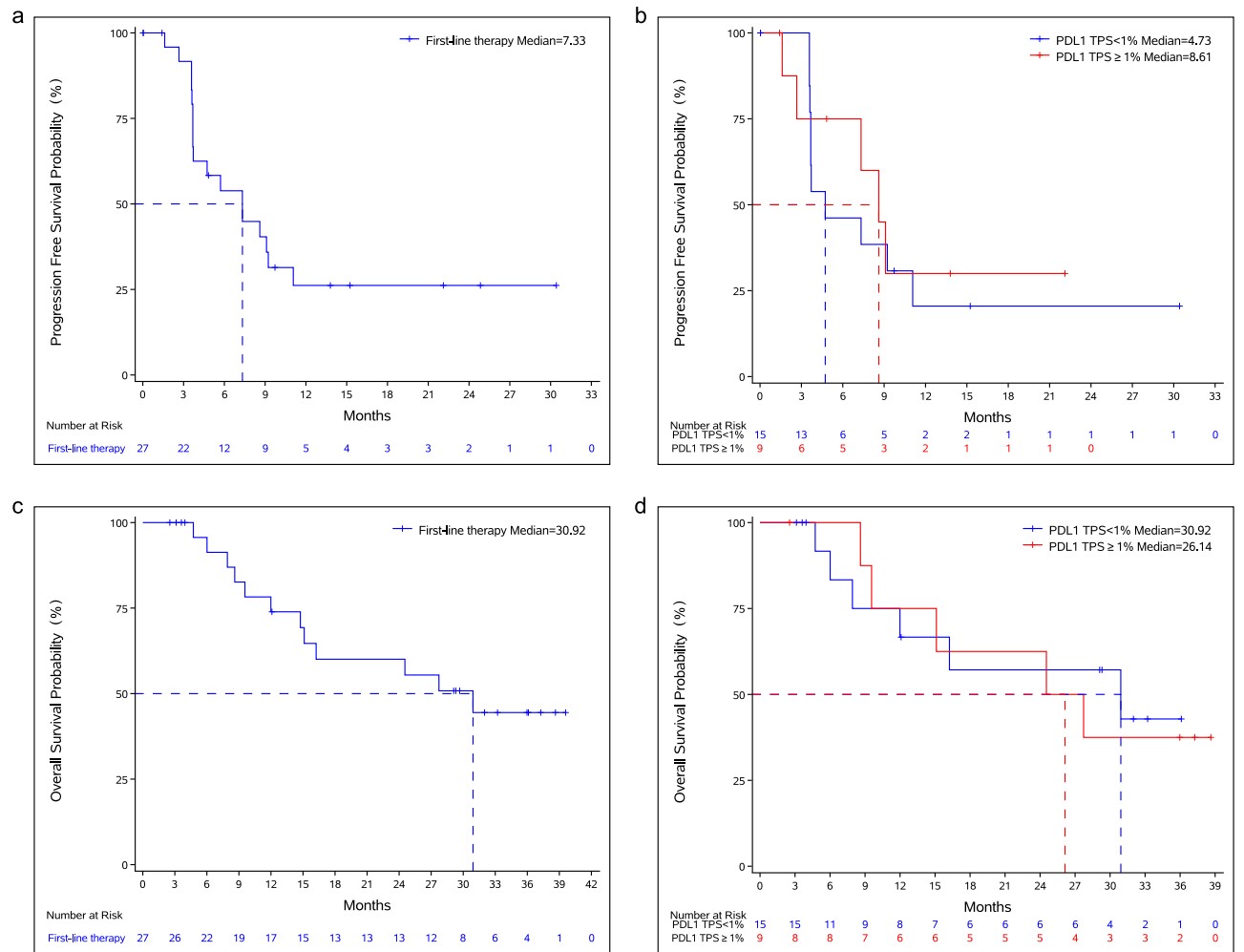

**Fig. 1 | Survival outcomes. a** Kaplan–Meier curve for progression-free survival. **b** Kaplan–Meier curves for progression-free survival based on PD-L1 expression status. **c** Kaplan–Meier curve for overall survival. **d** Kaplan–Meier curves for overall survival based on PD-L1 expression status. Source data are provided as a Source Data file.

**Table 3 | Survival outcomes assessed by IRC**

|  | PD-L1 TPS ≥ 1% | PD-L1 TPS < 1% | HR (95% CI) | *p*-Value | Total (*N* = 27) |
|---|---|---|---|---|---|
| Median PFS | 8.61 months | 4.73 months | 0.8 (0.27–2.35) | 0.6817 | 7.33 months |
| 95% CI | (1.61, -) | (3.61, 11.07) |  |  | (3.71–13.67) |
| Median OS | 26.14 months | 30.92 months | 1.1 (0.34–3.62) | 0.8717 | 30.92 months |
| 95% CI | (8.61, -) | (6.01, -) |  |  | (14.75, -) |
| 1-year OS rate | 75.0% | 66.7% |  |  | 73.9% |
| 95% CI | (31.5, 93.1) | (33.7, 86.0) |  |  | (50.9–87.3) |
| 2-year OS rate | 62.5% | 57.1% |  |  | 60.1% |
| 95% CI | (22.9, 86.1) | (25.4, 79.6) |  |  | (37.2–76.9) |

Survival analysis based on the Kaplan–Meier method was used for PFS and OS. A log-rank test was used to compare the difference in survival rate based on PD-L1 expression status. All *p*-values were two-sided.

*IRC* independent review committee, *HR* hazards ratio, *95% CI* 95% confidence interval, *PFS* progression-free survival, *OS* overall survival.

In the present multicenter, open-label, phase II study, we evaluated the efficacy and safety of KN046 plus nab-paclitaxel in the first-line treatment of locally advanced inoperable or metastatic TNBC patients. To our knowledge, this study represents the initial investigation assessing the use of bispecific antibodies as a first-line treatment for advanced TNBC, and the preliminary and final results of the present study have been reported in the American Association for Cancer Research (AACR) 2021 and the San Antonio Breast Cancer Symposium (SABCS) 2022 previously[18,19]. Consistent with the preliminary analysis[18], the final analysis showed favorable clinical efficacy and survival benefits with the treatment of KN046 plus nab-paclitaxel. With a median follow-up of 32.0 months, the reported overall efficacy and survival results were considered mature. The clinical efficacy reported was favorable with an ORR of 44.0% (95% CI: 24.4– 65.1%), and the favorable response was durable (median DoR of 13.3 months, IQR: 11.7 months, 23.2 months) among the response-evaluable patients regardless of PD-L1 expression status. Survival outcomes reported in this study indicate that unselected advanced

## Table 4 | Safety summary

| n (%) | KNO46 3 mg/kg Q2W + nab-paclitaxel (n = 16) | KNO46 5 mg/kg Q2W + nab-paclitaxel (n = 11) | Total (n = 27) |
|---|---|---|---|
| TEAE | 16 (100.0%) | 11 (100.0%) | 27 (100.0%) |
| TEAE associated with any study drug | 16 (100.0%) | 11 (100.0%) | 27 (100.0%) |
| TEAE Grade ≥ 3 | 11 (68.8%) | 7 (63.6%) | 18 (66.7%) |
| TEAE Grade ≥ 3 associated with any study drug | 11 (68.8%) | 7 (63.6%) | 18 (66.7%) |
| SAE | 6 (37.5%) | 3 (27.3%) | 9 (33.3%) |
| SAE associated with any study drug | 4 (25.0%) | 2 (18.2%) | 6 (22.2%) |
| irAE | 8 (50.0%) | 5 (45.5%) | 13 (48.1%) |
| irAE Grade ≥ 3 | 0 | 3 (27.3%) | 3 (11.1%) |
| Immune-related SAE | 0 | 1 (9.1%) | 1 (3.7%) |
| TEAE leading to any treatment withdrawal | 6 (37.5%) | 3 (27.3%) | 9 (33.3%) |
| TEAE leading to death | 2 (12.5%) | 0 | 2 (7.4%) |
| TEAE leading to death associated with KNO46 | 0 | 0 | 0 |

*TEAE* treatment-emergent adverse event, *SAE* serious adverse event, *irAE* immune-related adverse event.

## Table 5 | Treatment-related adverse events[a]

| n (%) | KNO46 3 mg/kg Q2W + nab-paclitaxel (n = 16) | | KNO46 5 mg/kg Q2W + nab-paclitaxel (n = 11) | | Total (n = 27) | |
|---|---|---|---|---|---|---|
| | Any Grade | Grade ≥3 | Any Grade | Grade ≥3 | Any Grade | Grade ≥3 |
| Any AE | 16 (100.0%) | 11 (68.8%) | 11 (100.0%) | 7 (63.6%) | 27 (100.0%) | 18 (66.7%) |
| Neutropenia | 12 (75.0%) | 7 (43.8%) | 6 (54.5%) | 2 (18.2%) | 18 (66.7%) | 9 (33.3%) |
| Leukopenia | 11 (68.8%) | 6 (37.5%) | 5 (45.5%) | 2 (18.2%) | 16 (59.3%) | 8 (29.6%) |
| Alopecia | 7 (43.8%) | 0 | 7 (3.6%) | 0 | 14 (51.9%) | 0 |
| Elevated ALT | 8 (50.0%) | 0 | 5 (45.5%) | 0 | 13 (48.1%) | 0 |
| Elevated AST | 7 (43.8%) | 2 (12.5%) | 6 (54.5%) | 1 (9.1%) | 13 (48.1%) | 3 (11.1%) |
| Pyrexia | 5 (31.3%) | 0 | 5 (45.5%) | 1 (9.1%) | 10 (37.0%) | 1 (3.7%) |
| Rash | 6 (37.5%) | 0 | 3 (27.3%) | 1 (9.1%) | 9 (33.3%) | 1 (3.7%) |
| Anemia | 6 (37.5%) | 0 | 2 (18.2%) | 0 | 8 (29.6%) | 0 |
| Elevated γ-GGT | 4 (25.0%) | 1 (6.3%) | 3 (27.3%) | 3 (27.3%) | 7 (25.9%) | 4 (14.8%) |
| Vomiting | 5 (31.3%) | 0 | 1 (9.1%) | 0 | 6 (22.2%) | 0 |
| Hypothyroidism | 3 (18.8%) | 0 | 3 (27.3%) | 0 | 6 (22.2%) | 0 |
| Hyponatremia | 4 (25.0%) | 2 (12.5%) | 1 (9.1%) | 0 | 5 (18.5%) | 2 (7.4%) |
| Hypokalemia | 5 (31.3%) | 4 (25.0%) | 0 | 0 | 5 (18.5%) | 4 (14.8%) |
| Anorexia | 4 (25.0%) | 1 (6.3%) | 1 (9.1%) | 0 | 5 (18.5%) | 1 (3.7%) |
| Diarrhea | 3 (18.8%) | 0 | 2 (18.2%) | 0 | 5 (18.5%) | 0 |
| Infusion reaction | 4 (25.0%) | 1 (6.3%) | 1 (9.1%) | 0 | 5 (18.5%) | 1 (3.7%) |
| Peripheral edema | 3 (18.8%) | 0 | 1 (9.1%) | 0 | 4 (14.8%) | 0 |
| Constipation | 4 (25.0%) | 0 | 0 | 0 | 4 (14.8%) | 0 |
| Elevated LDH | 3 (18.8%) | 0 | 0 | 0 | 3 (11.1%) | 0 |
| Elevated Bilirubin | 1 (6.3%) | 0 | 2 (18.2%) | 0 | 3 (11.1%) | 0 |
| Allergic dermatitis | 2 (12.5%) | 0 | 1 (9.1%) | 0 | 3 (11.1%) | 0 |
| Fatigue | 0 | 0 | 3 (27.3%) | 0 | 3 (11.1%) | 0 |
| Hypocalcemia | 3 (18.8%) | 0 | 0 | 0 | 3 (11.1%) | 0 |
| Hyperglycosemia | 3 (18.8%) | 1 (6.3%) | 0 | 0 | 3 (11.1%) | 1 (3.7%) |
| Infectious pneumonia | 3 (18.8%) | 2 (12.5%) | 0 | 0 | 3 (11.1%) | 2 (7.4%) |
| URI | 1 (6.3%) | 0 | 2 (18.2%) | 0 | 3 (11.1%) | 0 |
| Hypoesthesia | 2 (12.5%) | 0 | 1 (9.1%) | 0 | 3 (11.1%) | 0 |
| Ostealgia | 2 (12.5%) | 0 | 1 (9.1%) | 0 | 3 (11.1%) | 0 |

[a]Includes any-grade AEs that occurred in ≥10% of patients in either group.

*AE* adverse event, *ALT* alanine aminotransaminase, *AST* aspartate aminotransferase, *γ-GGT* γ-glutamyl transferase, *LDH* lactic dehydrogenase, *URI* upper respiratory infection.

TNBC patients benefit from the first-line treatment of KN046 plus nab-paclitaxel, with a median PFS of 7.33 months (95% CI: 3.68–11.07 months) and a median OS of 30.92 months (95% CI: 14.75–NE months) among the ITT population. Although the present study was only a phase II trial with limited sample size, and different studies could not be compared directly due to different study designs and populations, the survival outcomes achieved in the present study were still promising. Among advanced TNBC regardless of PD-L1 expression status, the survival outcomes in first-line treatment reported in this study were encouraging compared with that in KEYNOTE-355 trial (PFS = 7.5 months, OS = 17.2 months, pembrolizumab plus chemotherapy), IMpassion 130 trial (PFS = 7.2 months, OS = 21.0 months, atezolizumab plus nab-paclitaxel), and IMpassion 131 trial (PFS = 5.7 months, OS = 19.2 months, atezolizumab plus paclitaxel)[3–8,20]. In addition, among the PD-L1 positive advanced TNBC patients in the present study, the combination treatment of KN046 plus nab-paclitaxel has shown potentially superior PFS benefits compared to PD-L1 negative patients. In PD-L1 positive patients, PFS was 8.61 months (versus 4.73 months), and the 2-year OS rate was 62.5% (versus 57.1%) compared to PD-L1 negative patients. In the previous Phase III trials of ICI plus chemotherapy in first-line treatment of mTNBC patients, PFS in PD-L1 positive population was 7.6 months in the KEYNOTE-355 trial, 7.5 months in the IMpassion 130 trial, and 6.0 months in IMpassion 131 trial, respectively. In our study, no significant difference in clinical efficacy or survival outcomes was observed between the two dose-level groups of KN046 (DL-1: 3 mg/kg Q2W; DL-2: 5 mg/kg). The clinical benefit of KN046 plus nab-paclitaxel in the first-line treatment of mTNBC is worth validation in further clinical studies with larger sample sizes.

The safety profile of the combination treatment of KN046 plus nab-paclitaxel in this study was consistent with that reported for anti-PD-1/PD-L1 plus chemotherapy in advanced TNBC. The majority of TRAEs were hepatotoxicity and hematotoxicity, which were reversible after symptomatic and supportive treatment. The majority of irAEs were Grade 1/2, and the incidence of ≥Grade 3 irAEs was 11.1%, which is comparable with that in KEYNOTE-355 trial (5%, pembrolizumab plus chemotherapy), IMpassion 130 trial (8.7%, atezolizumab plus nab-paclitaxel), and IMpassion 131 trial (11%, atezolizumab plus paclitaxel)[4,6,7]. In addition, grade ≥3 irAEs and immune-related SAEs were only reported in the DL-2 group with higher dose of KN046, with no KN046 treatment-related adverse events (AEs) leading to death. Grade ≥3 irAEs were reported in 3 patients, 2 patients with Grade 3 immune-mediated liver disease and 1 patient with Grade 3 rash. Immune-related SAEs were reported in 1 patient with immune-mediated liver disease. As for the endocrine-related AEs reported in this study, hypothyroidism (3 of 27 patients, 11.1%) and hyperthyroidism (1 of 27 patients, 3.7%) occurred in patients treated with KN046 plus nab-paclitaxel, none of the patients developed diabetes mellitus or adrenal insufficiency. The incidence of endocrine-related AEs was comparable to even lower than that in the KEYNOTE-355 trial (pembrolizumab plus chemotherapy), IMpassion 130 trial (atezolizumab plus nab-paclitaxel), and IMpassion 131 trial (atezolizumab plus paclitaxel)[4,6,7]. Data from this study has shown good tolerance and management of KN046 plus nab-paclitaxel in the first-line treatment of advanced TNBC. Based on efficacy and toxicity profile, the 3 mg/kg Q2W dose level of KN046 in combination with nab-paclitaxel was associated with comparable efficacy and superior safety.

We acknowledge the limitations of the present study. Firstly, the sample size was small, and the study enrollment was restricted due to the pandemic. As we have demonstrated the promising efficacy and tolerance of KN046 plus nab-paclitaxel in the first-line treatment of advanced TNBC, the combination strategy is worth investigating in follow-up clinical studies with larger sample sizes. Data from this study has shown that the 3 mg/kg Q2W dose level of KN046 was associated with comparable efficacy and superior safety, but there was no statistical difference between the two dose groups due to the limited sample size. The appropriate dosage level of KN046 also needs further investigation in follow-up phase III trials. Moreover, there was an absence of a control group in the present study. The comparison of this combination strategy with standard treatment strategy or other immunotherapy plus chemotherapy in the first-line treatment of advanced TNBC needs to be evaluated in further randomized clinical studies. In addition, besides PD-L1 expression status, other predictive biomarkers for efficacy and toxicities need to be explored in further investigations.

In conclusion, KN046, in combination with nab-paclitaxel, showed favorable clinical efficacy and encouraging survival outcomes in the first-line treatment of locally advanced inoperable or metastatic TNBC, especially PD-L1 positive patients. Patients in this study tolerated the combination therapy well, and the safety profile was manageable. We look forward to further investigation in follow-up studies with a larger sample size, and we also expect that this immunotherapy agent can become a treatment option for TNBC patients.

## Methods

### Ethical statement

The present study (KN046-203) was approved by the ethics committee of the National Cancer Center/Cancer Hospital, Chinese Academy of Medical Sciences, and Peking Union Medical College (Approval No. 2018L02884) and was registered at clinicaltrials.gov (NCT03872791, www.clinicaltrials.gov) on March 13, 2019. The study design and conduct complied with all relevant regulations regarding the use of human study participants and was conducted in accordance with the criteria set by the Declaration of Helsinki. The study was performed in accordance with the relevant guidelines and regulations. All patients voluntarily participated and provided written informed consent before the initiation of any study-related treatment or procedures.

### Study design

The present study (KN046-203, NCT03872791) was a multicenter, open-label, phase II study with two treatment cohorts (KN046 monotherapy or in combination with Nab-paclitaxel in patients with Triple-negative Breast Cancer).

The monotherapy cohort aimed to evaluate the efficacy and safety of KN046 alone in locally advanced inoperable or metastatic TNBC patients who have failed at least one prior anthracycline and taxanes-containing systemic treatment, and the results will be reported separately.

Here, we present the results of the combination therapy cohort, aiming to evaluate the efficacy and safety of KN046 in combination with nab-paclitaxel in patients with treatment-naïve locally advanced inoperable or metastatic TNBC. Eligible patients received the treatment of KN046 at two dose levels (3 mg/kg Q2W or 5 mg/kg Q2W) plus nab-paclitaxel sequentially, with a dose escalation part and a dose expansion part afterward. Efficacy and safety assessments were conducted monthly until disease progression, intolerable AEs, withdrawal of informed consent, or KN046 treatment over 2 years. The study design is shown in Fig. 2.

The primary endpoints included ORR and DoR assessed by IRC. The secondary endpoints included ORR and DoR assessed by investigators, DCR and CBR assessed by IRC and investigators, PFS, 1-year/2-year OS rate, safety/tolerability, and the correlation of PD-L1 status with clinical efficacy. Due to the sample limitations, additional planned secondary endpoints, including immunogenicity, pharmacokinetics, additional biomarkers, and correlation between drug exposure levels and anti-tumor activity, were not fully performed. The trial protocol is available in the Supplementary Information file.

### Patients

The enrollment was conducted between May 30, 2019 and June 30, 2020. Women aged 18–70 years with histologically confirmed locally

**Fig. 2 | Study design.** Abbreviation: TNBC triple-negative breast cancer, TFI treatment-free interval, ECOG PS Eastern Cooperative Oncology Group performance status, DL dose level, ORR objective response rate, DoR duration of response, CBR clinical benefit rate, DCR disease control rate, PFS progression-free survival, OS overall survival.

advanced unresectable or metastatic TNBC were eligible. Additional inclusion criteria included: ≥18 years old, naïve system treatment for metastatic TNBC, prior chemotherapy in the neoadjuvant/adjuvant therapy including taxanes was allowed if treatment-free interval (TFI) ≥ 12 months, measurable disease at baseline according to RECIST v1.1, an Eastern Cooperative Oncology Group performance status (ECOG PS) of 0/1, and adequate organ functions.

Key exclusion criteria included untreated active central nervous system (CNS) metastasis or leptomeningeal metastasis; receiving immunosuppressive agents (such as steroids) for any reason; having interstitial lung disease or a history of pneumonitis that required oral or intravenous glucocorticoids to assist with management; having an active autoimmune disease that might deteriorate when receiving an immunostimulatory agent; history of uncontrolled intercurrent illness; or known severe hypersensitivity reactions to antibody drug.

The first 6 enrolled patients were enrolled in the 3 mg/kg group (dose Level [DL]-1 group). Firstly, The Site Monitoring Committee (SMC) meeting was held as all 6 subjects had completed the 28-day safety observation period. After an evaluation of the safety, initial efficacy, and pharmacokinetic data, we decided to increase to the 5 mg/kg group (DL-2 group) and also expand the 3 mg/kg group at the same time. The 5 mg/kg group first enrolled 6 patients, and after the evaluation of the SMC meeting, as all subjects had completed the 28-day safety observation period, we decided to expand the 5 mg/kg group.

### Therapeutic schedule
KN046 at the dose of 3 mg/kg or 5 mg/kg was administered intravenously on days 1, and 15 of every 28-day cycle, nab-paclitaxel at the dose of 100 mg/m$^2$ was administered intravenously on days 1, 8, and 15 of every 28-day cycle.

### Efficacy and safety assessment
Tumor response was evaluated every 8 weeks according to the Response Evaluation Criteria in Solid Tumors (RECIST) v1.1[21]. ORR was defined as the portion of patients who have achieved complete response (CR) and partial response (PR). DoR was defined as the time from the date of initial assessed CR or PR until the date of disease progression or death from any cause, whichever occurs first. DCR was defined as the portion of patients who have achieved CR, PR, and stable disease (SD). CBR was defined as the portion of patients who have achieved CR, PR, and SD ≥ 24 weeks. PFS was defined as the time from the first dose administration to the date of disease progression or cancer-related death, whichever occurs first. OS was defined as the time from the first dose administration to the date of death or last visit.

AEs were collected and graded according to the National Cancer Institute Common Terminology Criteria for Adverse Events (CTCAE) v5.0 assessed by IRC[22].

### PD-L1 expression
Each enrolled patient was required to provide tumor tissue samples for the determination of tumor and immune cell PD-L1 status. Immune cell PD-L1 expression was assessed by VENTANA PD-L1 (SP142) immunohistochemistry assay (Abcam anti-PD-L1 (SP142) Antibody, catalog number ab228462, dilution ratio 1:200, 0.4 μg/ml). PD-L1-positive was defined as PD-L1 expression ≥ 1% and PD-L1-negative was defined as PD-L1 expression <1%.

### Statistical analysis
This study used an electronic data acquisition system to create an electronic case report form (eCRF), which was logged online through the Internet for data collection and management. Statistical analyses were performed using SAS statistical analysis software (version 9.4). The 95% confidence interval (CI) was calculated using the Clopper Pearson method. Survival analysis based on the Kaplan–Meier method was used for DoR, PFS, and OS. The cumulative PFS rate and OS rate at each time point were measured in increments over 3 months, and the status of enrolled patients at 6 months, 12 months, and at the end of the study was measured. A log-rank test was used to compare the difference in survival rate based on PD-L1 expression status. All p-values were two-sided.

### Reporting summary
Further information on research design is available in the Nature Portfolio Reporting Summary linked to this article.

## Data availability
The raw clinical data are protected and are not available due to data privacy laws. The de-identified datasets supporting the findings of this study are available for academic purposes on request from the corresponding author (Binghe Xu) at any time with the approval of the Institutional Ethical Committees and will be available for 5 years. The trial protocol is available as Supplementary Note in the Supplementary Information. The remaining data are available within the Article, Supplementary Information, or Source Data file. Source data are provided as a Source Data file. Source data are provided in this paper.

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

## Acknowledgements

The study was funded by Jiangsu Alphamab Biopharmaceuticals Co., Ltd. and partly supported by CAMS Innovation Fund for Medical Sciences (CIFMS) 2021-I2M-1-014 (B.X.) and 2022-I2M-2-002 (B.X.). The authors and sponsor collaborated in data collection and assembly and guaranteed the authenticity and integrity of the data. The corresponding author prepared the initial draft of the manuscript with support from the sponsor. We thank all the patients, their families, all the investigators, and their institutions for the time and effort put into this study.

## Author contributions

Conception and design: Q.L. and B.X.; Provision of study material or patients: Q.L., Q.Z., Q.O., Y.Z., Q.L., T.S., F.Y., and B.C.Z.; Collection and assembly of data: J.L., S.X., and B.Y.Z.; Data analysis and interpretation: J.L.; Manuscript writing: Q.L. and J.L. Final paper editing: Q.L., J.L., and B.X.; Final approval of paper: All authors.

## Competing interests

B.X. reports receiving advisory fees from Novartis and AstraZeneca and fees for serving on a speakers' bureau from Pfizer and Roche. The remaining authors declare no competing interests.

## Additional information

[1]Department of Medical Oncology, National Cancer Center/National Clinical Research Center for Cancer/Cancer Hospital, Chinese Academy of Medical Sciences and Peking Union Medical College, Beijing 100021, China. [2]Harbin Medical University Cancer Hospital/Oncology Department, Harbin, Heilongjiang 150076, China. [3]Hunan Cancer Hospital, Changsha, Hunan 410031, China. [4]Liaocheng People's Hospital, Liaocheng, Shandong 252004, China. [5]Breast Tumor Center, Sun Yat-Sen Memorial Hospital, Sun Yat-Sen University, Guangzhou, Guangdong 510120, China. [6]Liaoning Cancer Hospital & Institute, Cancer Hospital of China Medical University, Shenyang, Liaoning 110801, China. [7]The First Affiliated Hospital of Xiamen University, Xiamen University, Xiamen, Fujian 361003, China. [8]Nantong Tumor Hospital, Nantong, Jiangsu 226006, China. [9]Jiangsu Alphamab Biopharmaceuticals Co., Ltd., Suzhou, Jiangsu 215127, China. [10]State Key Laboratory of Molecular Oncology, Department of Medical Oncology, National Cancer Center/National Clinical Research Center for Cancer/Cancer Hospital, Chinese Academy of Medical Sciences and Peking Union Medical College, Beijing 100021, China. ✉e-mail: xubinghe@medmail.com.cn

