## [Peer Review File · Nature Communications]

REVIEWER COMMENTS

Reviewer #1 - Breast cancer immunotherapy (Remarks to the Author):

Qiao Li et al. described a clinical phase II study of KN046 an anti-PD-L1/CTLA-4 bispecific antibody) in combination with nab paclitaxel in first-line treatment of metastatic triple-negative breast cancer. One of the highlights of this study is the first exploration of bispecific antibody combined with chemotherapy in the first-line treatment of metastatic TNBC patients. Nevertheless, the design of the trial is basically "ICB+chemotherapy", which is not very different or unique from current treatment available.

Another concern is that the sample size is too small ($n = 27$), and this sample is further divided into two groups. There is not enough statistical power to answer the efficacy and safety questions of the treatment scheme, which can only lead to a descriptive analysis. Furthermore, this is a multi-center study consists of 9 different centers, which should have larger cohort from each center.

Second, the heterogeneity of mTNBC patients is large, Therefore, it is very important to accurately find the benefit group of immunotherapies. This study did not explore biomarkers specific to TNBC or mTNBC, therefore it failed to find the dominant-beneficial group.

Reviewer #2 - Biostatistics and clinical trials (Remarks to the Author):

Overview:

In this multicenter, open-label, phase II study, a total of 27 patients (histologically confirmed locally advanced, unresectable or metastatic TNBC) were randomly assigned to receive an anti-PD-L1/CTLA-4 bispecific antibody, KN046, at a dose level of 3 mg/kg (16 patients) or 5 mg/kg (11 patients) in combination with nab-paclitaxel, to evaluate the efficacy and safety of the anti-PD-L1/CTLA-4 bispecific antibody KN046 in combination with nab-paclitaxel in the first-line treatment of patients with locally advanced, unresectable or metastatic TNBC.

The primary endpoints were objective response rate (ORR) and duration of response (DoR) as assessed by an independent review committee (IRC). Secondary endpoints included investigator-assessed ORR and DoR, IRC- and investigator-assessed disease control rate (DCR) and clinical benefit rate (CBR), PFS, 1-year/2-year OS rate, and safety/tolerability.

This trial was registered on clinicaltrials.gov (NCT03872791).

My review will focus on the statistical aspects (study design and data analysis) of this study due to my background.

Objective:

The objective of this study is to evaluate the efficacy (ORR, DoR, PFS, OS, ...) and safety of the anti-PD-L1/CTLA-4 bispecific antibody KN046 in combination with nab-paclitaxel in the first-line treatment of locally advanced unresectable or metastatic TNBC patients regardless of PD-L1 status.

Study design and statistical analysis:

The study was a multicenter, randomized, open-label, Phase II trial designed to evaluate the efficacy and safety of KN046 in combination with nab-paclitaxel in patients with treatment-naïve locally advanced inoperable or metastatic TNBC. Eligible patients received KN046 at two dose levels (3mg/kg Q2W or 5mg/kg Q2W) plus nab-paclitaxel, and efficacy and safety were evaluated monthly until disease progression, intolerable adverse events (AEs), withdrawal of consent, or discontinuation of KN046 for 2 years.

Statistical analyses were performed using SAS statistical analysis software (version 9.4). The 95% confidence interval (CI) for response rates was calculated using the Clopper-Pearson method. Kaplan-Meier survival analysis was used for DoR, PFS, and OS. The cumulative PFS rate and OS rate at each time point were measured in increments over 3 months, and the status of enrolled patients at 6 months, 12 months, and at the end of the study were measured. The log-rank test was used to compare the difference in survival based on PD-L1 expression status. All p-values were two-tailed.

Main Results:

27 patients were enrolled from July 2019 to July 2020, with 16 patients randomly assigned to the 3 mg/kg dose and 11 patients randomly assigned to the 5 mg/kg dose. Median follow-up was 32.0 months.

Responses:

The ORR (among the 25 patients evaluable for response) was 44.0% (95% CI, 24.4% -65.1%), and the median DoR was 13.3 months (IQR: 11.9, 23.2).

PFS:

In the intention-to-treat (ITT) population, median PFS reached 7.33 months (95%CI, 3.68 - 11.07 months) and median PFS reached 8.61 months (95%CI, 1.61 - NE months) in PD-L1 positive patients.

OS:

The median OS was 30.92 months (95%CI, 14.75 - NE months) and the 2-year OS rate was 60.1% (95%CI, 37.2% - 76.9%).

76.9%). The median OS was 26.14 months (95%CI, 8.61 - NE months) and the 2-year OS rate was 62.5% (95%CI, 22.9% - 86.1%) in PD-L1 positive patients.

Safety:

Patients tolerated the combination therapy well. The incidence of grade ≥ 3 treatment-emergent adverse events (TEAEs) was 66.7% and the incidence of serious adverse events (SAEs) was 33.3%, with no KN046 treatment-emergent TEAEs leading to death.

Comments:

Overall, this paper is well written. Most aspects of the design and statistical analysis are clearly described. The results look promising, but I will not comment on the clinical/scientific aspects of this study. I do have a few minor questions/comments for the authors to consider.

Regarding randomization:

It might be appropriate to mention that this is a randomized trial, e.g. "The study was a multicenter, randomized, open-label, phase II trial ...".

It would be helpful to provide more details about the randomization plan, such as how random assignments were generated, how many sites were included, and whether randomization was performed within sites.

In addition, how could there be an imbalance in the number of patients (16 vs. 11) in the two dose levels?

Blindness:

Although this is an open-label study, it is still important to clarify whether the investigators who assessed the endpoints (efficacy, safety) were blinded to the treatment assignment of the patients.

Analysis

It is mentioned that "The 95% confidence interval (CI) was calculated using the Clopper-Pearson method". This may not be accurate. The Clopper-Pearson method constructs CIs for binary outcomes such as **response rate**, but not for time-to-event outcomes such as OS/PFS.

In comparing LD-1 positive vs. negative for PFS/OS, did you control for dose level?

Others:

Seems like there were 3 patients with unknown PD-L1 status. What happened here?

Reviewer #3 - Breast cancer immunotherapy (Remarks to the Author):

This is an interesting manuscript reporting on a Phase II clinical trial (NCT03872791). The findings are interesting and novel in that KN046 is a new anti-PD-L1/CTLA-4 bispecific antibody. I have a few questions/suggestions for your consideration:

- 1) Was the trial stopped due to the pandemic? If so, please state the planned versus the actual accrual. Please state any dates of interruption of enrollment.
- 2) What was the role of the sponsor in activities such as data analysis, manuscript preparation, decision to publish?
- 3) How do rates of endocrine adverse effects compare to pembrolizumab?
- 4) Please elaborate on what trials should happen next for KN046.
- 5) What dose would you select for the next trial?
- 6) Although the manuscript is generally written in good English, a few exceptions are noted. Please try to improve the English writing for reader comprehension.
- 7) On clinicaltrials.gov it states the trial will also evaluate Percentage of subjects with anti-drug antibodies. Was this done?

RESPONSE TO REVIEWERS' COMMENTS

Reviewer #1 - Breast cancer immunotherapy (Remarks to the Author):

Qiao Li et al. described a clinical phase II study of KN046 (an anti-PD-L1/CTLA-4 bispecific antibody) in combination with nab paclitaxel in first-line treatment of metastatic triple-negative breast cancer. One of the highlights of this study is the first exploration of bispecific antibody combined with chemotherapy in the first-line treatment of metastatic TNBC patients. Nevertheless, the design of the trial is basically "ICB+chemotherapy", which is not very different or unique from current treatment available.

Response:

Thanks for your careful reading and meaningful comments! As you mentioned, we admit that the treatment strategy of "ICB+chemotherapy" in the present trial is a basic design. Although pembrolizumab in combination with chemotherapy may have reformed the choice of first-line treatment for advanced TNBC according to Keynote-355 trial^[1], the results of immunotherapy combined with chemotherapy in advanced TNBC are frequently unsatisfactory. It is still urgent to explore better synergistic treatment strategies of immunotherapy and chemotherapy in the treatment of advanced TNBC.

The combination of KN046 with chemotherapy has shown promising efficacy and well-tolerated toxicity in different advanced solid tumors including non-small cell lung cancer (NSCLC)^[2], pancreatic cancer^[3], esophageal squamous cell carcinoma^[4] and rare thoracic tumors^[5].

However, in the field of breast cancer, it is the first study reported to explore the efficacy and safety profile of this unique bispecific antibody combined with chemotherapy for advanced TNBC patients, especially in the first-line treatment. The good results of this small-size trial were encouraging for us to conduct follow-up studies with a larger sample size, and we also expect that this novel immunotherapy agent can become a new treatment option for TNBC patients.

Another concern is that the sample size is too small (n = 27), and this sample is further divided into two groups. There is not enough statistical power to answer the efficacy and safety questions of the treatment scheme, which can only lead to a descriptive analysis. Furthermore, this is a multi-center study consists of 9 different centers, which should have larger cohort from each center.

Response:

As you mentioned, we cannot deny that the sample size is too small, we also discussed the limitations of the present study, including the absence of a control group and limited number of enrolled patients in the discussion section. Per design, the present study is a phase II study, there was a dose escalation part and a dose expansion part afterward. One of the main objectives is to explore and evaluate the better dose group for conducting follow-up studies, and the 3mg/kg Q2W dose level of KN046 in combination with nab-paclitaxel was associated with comparable efficacy and superior

safety.

Per design, The combined treatment cohort will first enroll 6 subjects for 3mg/kg dose group as required by the protocol. The SMC meeting will be held after all 6 subjects are enrolled and all have completed the 28-day safety observation period. Review of the safety, initial efficacy, and pharmacokinetic data decided to: 1) Continue to expand to 25 subjects in the 3 mg/kg dose group; And/or 2) increased to the 5 mg/kg dose group. The 5 mg/kg dose group will first enroll 6 subjects, and the SMC meeting will be held after all 6 subjects are enrolled and all have completed the 28-day safety observation period. To review the safety, initial efficacy, and pharmacokinetic data of KN046 in combination with albumin-paclitaxel to determine whether to continue expansion to 25 subjects in the combined 5 mg/kg dose group. Thus, we planned to enroll 25-50 patients in total, and as you mentioned, this is a multi-center study consisting of 9 different centers. However, during the enrollment period from 2019 to 2020, the pandemic has had a certain degree of impact on the overall study enrollment plan and program implementation. At the height of the outbreak, hospitals were unable to treat patients, clinical research was restricted, and many patients were unable to access hospitals for treatment due to the pandemic. In the later stage of the study, the sample size was inevitably limited and 27 patients actually enrolled.

The present study is indeed a very preliminary and early exploratory study, but with promising efficacy and safety data achieved for the first-line treatment of advanced TNBC patients. The combination strategy is worth investigating in follow-up clinical studies with larger sample size, it is also hoped that the current data will serve as a preliminary accumulation for the follow-up studies.

Second, the heterogeneity of mTNBC patients is large, Therefore, it is very important to accurately find the benefit group of immunotherapies. This study did not explore biomarkers specific to TNBC or mTNBC, therefore it failed to find the dominant-beneficial group.

Response:

Thanks for your comments! As you mentioned, the heterogeneity of mTNBC patients is large and clinicians are also exploring different subtypes of TNBC in recent years. As we designed, we hope to collect blood and tissue samples from enrolled patients for biomarker exploration to find the advantageous populations. Unfortunately, due to the impact of the pandemic, samples of some patients were not available. Thus in the present study we only explored the PD-L1 expression status, as we mentioned the limitations in the last paragraph of the discussion section, other predictive biomarkers for efficacy and toxicities need to be explored in further investigations.

Special thanks to you for your constructive comments! We hope this revise meets with approval.

Reviewer #2 - Biostatistics and clinical trials (Remarks to the Author):

Please see the attached report.

Response:

Thanks for your careful reading again! We appreciate that you summarized our study design and results in the attached report and the valuable comments you mentioned. We answered your questions point by point as listed below, we hope you could be satisfied with the revisions.

Comments:

Regarding randomization:

It might be appropriate to mention that this is a randomized trial, e.g. "The study was a multicenter, randomized, open-label, phase II trial ...". It would be helpful to provide more details about the randomization plan, such as how random assignments were generated, how many sites were included, and whether randomization was performed within sites.

In addition, how could there be an imbalance in the number of patients (16 vs. 11) in the two dose levels?

Response:

This study is not a randomized study, but a prospective study. Per design, there was a dose escalation part and a dose expansion part afterwards. Patients were allocated to 3mg/kg dose group and 5mg/kg dose group without a randomization process.

Blindness:

Although this is an open-label study, it is still important to clarify whether the investigators who assessed the endpoints (efficacy, safety) were blinded to the treatment assignment of the patients.

Response:

This study is non-blind. Investigators who assess the safety were not blinded to the study treatment. While the primary endpoint of ORR and DoR were assessed by BICR who were blinded to the treatment allocation.

Analysis

It is mentioned that "The 95% confidence interval (CI) was calculated using the Clopper-Pearson method". This may not be accurate. The Clopper-Pearson method constructs CIs for binary outcomes such as response rate, but not for time-to-event outcomes such as OS/PFS.

Response:

In the statistical analyses of this study, the 95% confidence interval (CI) for response rates (ORR) was calculated using the Clopper-Pearson method. Kaplan-Meier survival analysis was used for time to event outcomes such as DoR, PFS, and OS. If you have any further questions or comments, please let me know, thanks a lot!

In comparing PD-L1 positive vs. negative for PFS/OS, did you control for dose level?

Response:

We are sorry that the dose level was not included in the Cox regression model in comparing PD-L1 positive vs. negative for PFS/OS. Besides, we have analyzed the survival outcomes between the two dose groups and also PFS/OS comparing PD-L1 expression levels between the two groups. We have added this part of results marked in red in the Results section and presented as Supplementary Table 2, hope it meets with approval!

Others:

Seems like there were 3 patients with unknown PD-L1 status. What happened here?

Response:

There is a tissue detachment happened in their histological samples, the central lab cannot get the PD-L1 result, therefore there were 3 patients with unknown PD-L1 status. We are sorry that we didn't explain the reason why these patients with unknown PD-L1 status in the baseline characteristics section of the patients, and we have added the reason in the first "Patients" section of the results section marked in red. Thanks again for your meaningful comment, hope this revise meet with approval!

Special thanks to you for your positive and constructive comments! We hope this revise meets with approval.

Reviewer #3 - Breast cancer immunotherapy (Remarks to the Author):

This is an interesting manuscript reporting on a Phase II clinical trial (NCT03872791). The findings are interesting and novel in that KN046 is a new anti-PD-L1/CTLA-4 bispecific antibody. I have a few questions/suggestions for your consideration:

Response:

Thanks for your careful reading and valuable comments again! We answered your questions point by point as listed below, we hope you could be satisfied with the revisions.

1) Was the trial stopped due to the pandemic? If so, please state the planned versus the actual accrual. Please state any dates of interruption of enrollment.

Response:

This trial began in May 2019 and ended in November 2022, there were 27 treatment-naïve locally advanced inoperable or metastatic TNBC patients enrolled to receive the treatment of KN046 plus nab-paclitaxel. Per design, The combined treatment cohort will first enroll 6 subjects for 3mg/kg dose group as required by the protocol. The SMC meeting will be held after all 6 subjects are enrolled and all have completed the 28-day safety observation period. Review of the safety, initial efficacy, and pharmacokinetic data decided to: 1) Continue to expand to 25 subjects in the 3 mg/kg dose group; And/or 2) increased to the 5 mg/kg dose group. The 5 mg/kg dose group will first enroll 6 subjects, and the SMC meeting will be held after all 6 subjects are enrolled and all have completed the 28-day safety observation period. To review the safety, initial efficacy, and pharmacokinetic data of KN046 in combination with albumin-paclitaxel to determine whether to continue expansion to 25 subjects in the combined 5 mg/kg dose group.

Thus, we planned to enroll 25-50 patients, and finally 27 patients were actually enrolled. As you mentioned, we cannot deny that the impact of the epidemic has had a certain degree of impact on the overall study enrollment plan and program implementation. At the height of the outbreak, hospitals were unable to treat patients, clinical research was restricted. According to the record of SMC meetings, patient enrollment was suspended on 2020/2/24 and 2020/4/27. And many patients were unable to access hospitals for treatment due to the pandemic. We also added it in the last paragraph of the discussion section, hope it meets with approval!

2) What was the role of the sponsor in activities such as data analysis, manuscript preparation, decision to publish?

Response:

As described in the “Author contributions” section, the sponsor (S.X. and B.Y.Z) contributed to data analysis and interpretation, collection and assembly. The decision to publish was approved by all authors including the sponsor.

3) How do rates of endocrine adverse effects compare to pembrolizumab?

Response:

Thank you for your comments. We are sorry that we didn't describe the endocrine adverse effects of this study strategy compared to pembrolizumab and other anti-PD-1/PD-L1 plus chemotherapy in advanced TNBC reported previously. Endocrine-related adverse effects are a kind of adverse reaction that needs special attention during the treatment of immunotherapy, we appreciate your constructive comments and we have added this content in the Results and Discussion sections marked in red. Hope this revise meets with approval!

As we described in "Safety and Tolerability" of the Results section, and the safety profile presented in table 4 and supplementary table 3. Immune-related adverse events (irAEs) were reported in 13 (48.1%) patients. Most irAEs were Grade 1 or 2. One of the most reported irAEs was hypothyroidism (3 patients, 11.1%). For other endocrine irAEs, hyperthyroidism occurred in 1 of 27 patients (3.7%), none of the patients developed diabetes mellitus or adrenal insufficiency.

When compared with pembrolizumab, as reported in the KEYNOTE-355 trial^[1], the incidence of irAEs in pembrolizumab–chemotherapy group was 26%, hypothyroidism occurred in 15% of patients, hyperthyroidism occurred in 5% of patients. When compared with atezolizumab, as reported in the IMpassion 130^[6] and IMpassion 131^[7] trials, irAEs occurred in 62% of patients who received atezolizumab plus paclitaxel and in 58.7% of patients who received atezolizumab plus nab-paclitaxel. The incidence of hypothyroidism was 18.3% (IMpassion 130, atezolizumab plus nab-paclitaxel) and 14% (IMpassion 131, atezolizumab plus paclitaxel). The incidence of hyperthyroidism was 4.8% (IMpassion 130, atezolizumab plus nab-paclitaxel) and 6% (IMpassion 131, atezolizumab plus paclitaxel). The incidence of diabetes mellitus was 1% (IMpassion 131, atezolizumab plus paclitaxel). The incidence of adrenal insufficiency was 1.1% (IMpassion 130, atezolizumab plus nab-paclitaxel) and 0.7% (IMpassion 131, atezolizumab plus paclitaxel).

Thus, the incidence of endocrine irAEs with the treatment of KN046 plus nab-paclitaxel in this study was comparable to or even lower than that reported in the previous trials. We cannot deny that the present study was a phase II study with limited samples, however, the current safety profile of endocrine irAEs was better, and we look forward to its performance in further studies.

4) Please elaborate on what trials should happen next for KN046.

Response:

In future trials, we tend to continue to conduct randomized controlled trials to investigate the efficacy and safety of KN046 in combination with chemotherapy against Pembrolizumab in combination with chemotherapy or against chemotherapy alone in the first-line treatment of advanced TNBC.

5) What dose would you select for the next trial?

Response:

As the efficacy, survival outcomes and toxicity profiles we described in the Results and Discussion section, the 3mg/kg Q2W dose level of KN046 in combination with nab-paclitaxel was associated with comparable efficacy and superior safety, which is worth

evaluating in the next trial.

Besides, we are sorry that the survival outcomes comparing the two dose groups were not presented in the previous manuscript. We have added this part of results marked in red in the Results section and presented as Supplementary Table 2, hope it meets with approval!

6) Although the manuscript is generally written in good English, a few exceptions are noted. Please try to improve the English writing for reader comprehension.

Response:

Thank you for your suggestion! An extensive proofreading was done, and our manuscript's English has now been improved.

7) On clinical trials.gov it states the trial will also evaluate Percentage of subjects with anti-drug antibodies. Was this done?

Response:

The titer of KN046 with anti-drug antibody (ADA) was evaluated as we stated as secondary measured outcomes on clinical trials.gov, 26 of 27 patients were evaluated. 14 of 26 (53.8%) patients with a negative ADA titer at baseline and after treatment. 10 of 26 (38.5%) patients with a positive ADA titer at baseline and a negative ADA titer after treatment. 1 of 26 (3.8%) patients with a positive ADA titer at baseline and a higher ADA titer (elevated less than 4 times) after treatment. None of patients with a positive ADA titer baseline and a higher ADA titer (elevated more than 4 times) after treatment.

This part of content was a description analysis, and we didn't present it in the current manuscript. Further exploration on ADA and immunogenicity of KN046 was ongoing and was also needed in future phase III trials.

Special thanks to you for your positive and constructive comments! We hope this revise meets with approval.

References:

- [1] CORTES J, CESCO D W, RUGO H S, et al. Pembrolizumab plus chemotherapy versus placebo plus chemotherapy for previously untreated locally recurrent inoperable or metastatic triple-negative breast cancer (KEYNOTE-355): a randomised, placebo-controlled, double-blind, phase 3 clinical trial [J]. *Lancet*, 2020, 396(10265): 1817-28.
- [2] ZHOU C, XIONG A, LI W, et al. P77.03 A Phase II Study of KN046 (Bispecific Anti-PD-L1/CTLA-4) in Patients (pts) with Metastatic Non-Small Cell Lung Cancer (NSCLC) [J]. *Journal of Thoracic Oncology*, 2021, 16(3): S636.
- [3] JIN G, GUO S, ZHANG Y, et al. Efficacy and safety of KN046 plus nab-paclitaxel/gemcitabine as first-line treatment for unresectable locally advanced or metastatic pancreatic ductal adenocarcinoma (PDAC) [J]. *Journal of Clinical Oncology*, 2021, 39(15_suppl): 4138-.
- [4] XU J, LIU R, ZHANG Y, et al. Efficacy and safety of KN046 plus paclitaxel/cisplatin as first-line treatment for unresectable locally advanced, recurrent or metastatic esophageal squamous cell carcinoma (ESCC) [J]. *Journal of Clinical Oncology*, 2021, 39(15_suppl): 4062-.
- [5] RICHARDSON G, KICHENADASSE G, GANJU V, et al. MA06.09 Preliminary Safety, Efficacy Results of KN046 (Bispecific Anti-PD-L1/CTLA4) in Subjects With Rare Thoracic Tumors [J]. *Journal of Thoracic Oncology*, 2021, 16(3): S154-S5.
- [6] EMENS L A, ADAMS S, BARRIOS C H, et al. First-line atezolizumab plus nab-paclitaxel for unresectable, locally advanced, or metastatic triple-negative breast cancer: IMpassion130 final overall survival analysis [J]. *Ann Oncol*, 2021, 32(8): 983-93.
- [7] MILES D, GLIGOROV J, ANDRÉ F, et al. Primary results from IMpassion131, a double-blind, placebo-controlled, randomised phase III trial of first-line paclitaxel with or without atezolizumab for unresectable locally advanced/metastatic triple-negative breast cancer [J]. *Ann Oncol*, 2021, 32(8): 994-1004.

REVIEWERS' COMMENTS

Reviewer #2 (Remarks to the Author):

Thanks for the responses to address my questions/comments.

On page 4 lines 169-172, it is stated that patients were assigned to two dose levels randomly. If this is not the case, then please remove the word "randomly" and provide a description about how patients were assigned to the two levels, e.g., sequentially, the first 16 to dose level 1 and the remaining 11 to dose level 2.

I have no other questions.

Reviewer #3 (Remarks to the Author):

The authors have appropriately addressed this reviewer's critiques. It is a minor point that there still remain a few issues with English language that could be addressed.

Noteworthy results: the author's data support their conclusion KN046 in combination with nab-paclitaxel showed favorable efficacy and survival benefits in the first-line treatment of locally advanced inoperable or metastatic TNBC patients, especially PD-L1 positive, with a tolerable and manageable toxicity profile.

Will the work be of significance to the field and related fields? This new drug KN046 shows promise and should be evaluated in larger, randomized clinical trials.

Is the methodology sound? Yes

Is there enough detail provided in the methods for the work to be reproduced? Yes

RESPONSE TO REVIEWERS' COMMENTS

Reviewer #2 (Remarks to the Author):

Thanks for the responses to address my questions/comments.

On page 4 lines 169-172, it is stated that patients were assigned to two dose levels randomly. If this is not the case, then please remove the word "randomly" and provide a description about how patients were assigned to the two levels, e.g., sequentially, the first 16 to dose level 1 and the remaining 11 to dose level 2.

I have no other questions.

Response:

Thanks again for your careful reading and constructive comment! We are sorry that there was a clerical error when describing how patients were assigned to the two levels.

There was a dose escalation part and a dose expansion part afterward. Per design, the combined treatment cohort will first enroll 6 subjects for 3mg/kg dose group as required by the protocol. The Site Monitoring Committee (SMC) meeting will be held after all 6 subjects are enrolled and all have completed the 28-day safety observation period. A review of the safety, initial efficacy, and pharmacokinetic data decided to: 1) Continue to expand to 25 subjects in the 3 mg/kg dose group; And/or 2) increase to the 5 mg/kg dose group. The 5 mg/kg dose group will first enroll 6 subjects, and the SMC meeting will be held after all 6 subjects are enrolled and all have completed the 28-day safety observation period. To review the safety, initial efficacy, and pharmacokinetic data of KN046 in combination with albumin-paclitaxel to determine whether to continue expansion to 25 subjects in the combined 5 mg/kg dose group.

In the actual completion process of the present study, 6 patients were enrolled to receive KN046 3mg/kg Q2W plus nab-paclitaxel (Dose Level [DL]-1 group) firstly, after the review of the SMC meeting, decided to increase to 5mg/kg group (DL-2 group) and also expand the 3mg/kg group at the same time. And the 5mg/kg group firstly enrolled 6 patients, after the review of the SMC meeting, decided to expand the 5mg/kg group. Finally, in the later stage of the study, the sample size was inevitably limited and 27 patients actually enrolled. So 16 patients were assigned to receive KN046 3mg/kg, and 11 patients were assigned to receive KN046 5mg/kg.

We are sorry that the description in the previous manuscript was not clear and was confusing, we have rephrased and modified the detailed patients' enrollment situation both in the methods and results sections, which were marked in red in the manuscript. And for the "randomly" you mentioned on page 4 lines 169-172, we have modified the description into "16 patients were assigned to receive KN046 3mg/kg Q2W plus nab-paclitaxel (DL-1 group) and the other 11 patients were assigned to receive KN046 5mg/kg Q2W plus nab-paclitaxel (DL-2 groups) sequentially, following a dose escalation part and a dose expansion part afterward."

Special thanks to you for your constructive comments! We hope this revise meets with

approval.

Reviewer #3 (Remarks to the Author):

The authors have appropriately addressed this reviewer's critiques. It is a minor point that there still remain a few issues with English language that could be addressed.

Response:

Thank you for your suggestion! An extensive proofreading was done, and our manuscript's English has now been improved.

Noteworthy results: the author's data support their conclusion KN046 in combination with nab-paclitaxel showed favorable efficacy and survival benefits in the first-line treatment of locally advanced inoperable or metastatic TNBC patients, especially PD-L1 positive, with a tolerable and manageable toxicity profile.

Will the work be of significance to the field and related fields? This new drug KN046 shows promise and should be evaluated in larger, randomized clinical trials.

Response: KN046 was the world's first recombinant humanized anti-PD-L1/CTLA-4 bispecific antibody, which was designed to achieve the synergistic effect of simultaneous blockade of PD-L1/PD-1 and CTLA-4 pathways. Preclinical and clinical studies have shown promising efficacy and tolerable toxicities in the advanced treatment of a variety of solid tumors.

The present study is the first study reported to evaluate the bispecific antibody in the first-line treatment of advanced TNBC, and the preliminary and final results of the present study have been reported in AACR 2021 and SABCS 2022 previously. The efficacy, survival, and safety results from the present study have shown that KN046 combined with nab-paclitaxel showed favorable efficacy and survival benefits with tolerable toxicity in the first-line treatment of metastatic TNBC, especially PD-L1 positive patients. The 3mg/kg Q2W dose level of KN046 in combination with nab-paclitaxel was associated with comparable efficacy and superior safety, we look forward to further investigation in follow-up studies with a larger sample size, and we also expect that this novel immunotherapy agent can become a new treatment option for TNBC patients.

Is the methodology sound? Yes

Response: The detailed methodology was described in the methods section and the source data was also provided in supplementary materials. If there was any confusion or question about the methodology, please feel free to contact us!

Is there enough detail provided in the methods for the work to be reproduced? Yes

Response: The detailed methodology was described in the methods section and the source data was also provided in supplementary materials. If there was any confusion or question about the methodology, please feel free to contact us!

Special thanks to you for your constructive comments! We hope this revise meets with approval.